# Extensive Diversity in *Escherichia coli* Group 3 Capsules Is Driven by Recombination and Plasmid Transfer from Multiple Species

Yaoqin Hong,[a,b] Jilong Qin,[a] Xavier Bertran Forga,[a] Makrina Totsika[a,b]

aCentre for Immunology and Infection Control, School of Biomedical Sciences, Queensland University of Technology, Queensland, Australia
bMax Planck Queensland Centre, Queensland University of Technology, Queensland, Australia

**ABSTRACT** Bacterial capsules provide protection against environmental challenges and host immunity. Historically, *Escherichia coli* K serotyping scheme, which relies on the hypervariable capsules, has identified around 80 K forms that fall into four distinct groups. Based on recent work by us and others, we predicted that *E. coli* capsular diversity is grossly underestimated. We exploited group 3 capsule gene clusters, the best genetically defined capsule group in *E. coli*, to analyze publicly available *E. coli* sequences for overlooked capsular diversity within the species. We report the discovery of seven novel group 3 clusters that fall into two distinct subgroups (3A and 3B). The majority of the 3B capsule clusters were found on plasmids, contrary to the defining feature of group 3 capsule genes localizing at the *serA* locus on the *E. coli* chromosome. Other new group 3 capsule clusters were derived from ancestral sequences through recombination events between shared genes found within the serotype variable central region 2. Intriguingly, flanking regions 1 and 3, known to be conserved areas among capsule clusters, showed considerable intra-subgroup variation in clusters from the 3B subgroup, containing genes of shared ancestry with other Enterobacteriaceae species. Variation of group 3 *kps* clusters within dominant *E. coli* lineages, including multidrug-resistant pathogenic lineages, further supports that *E. coli* capsules are undergoing rigorous change. Given the pivotal role of capsular polysaccharides in phage predation, our findings raise attention to the need of monitoring *kps* evolutionary dynamics in pathogenic *E. coli* in supporting phage therapy.

**IMPORTANCE** Capsular polysaccharides protect pathogenic bacteria against environmental challenges, host immunity, and phage predations. The historical *Escherichia coli* K typing scheme, which relies on the hypervariable capsular polysaccharide, has identified around 80 different K forms that fall into four distinct groups. Taking advantage of the supposedly compact and genetically well-defined group 3 gene clusters, we analyzed published *E. coli* sequences to identify seven new gene clusters and revealed an unexpected capsular diversity. Genetic analysis revealed that group 3 gene clusters shared closely related serotype-specific region 2 and were diversified through recombination events and plasmid transfer between multiple Enterobacteriaceae species. Overall, capsular polysaccharides in *E. coli* are undergoing rigorous change. Given the pivotal role capsules play in phage interactions, this work highlighted the need to monitor the evolutionary dynamics of capsules in pathogenic *E. coli* for effective phage therapy.

**KEYWORDS** capsular polysaccharide, surface polysaccharide, evolution, *Escherichia coli*, phage receptor, glycobiology, evolutionary biology, polysaccharides

Address correspondence to Yaoqin Hong, yaoqin.hong@qut.edu.au, or Makrina Totsika, makrina.totsika@qut.edu.au.

The authors declare no conflict of interest.

The cell surface of many *Escherichia coli* strains is enveloped in a gelatinous layer composed of tightly packed strands of long-chain polysaccharides called capsular polysaccharide or capsule (1). *E. coli* capsules are structurally diverse: over 80 capsular serotypes are known to date and referred to as different K antigens in the *E. coli* serotyping scheme

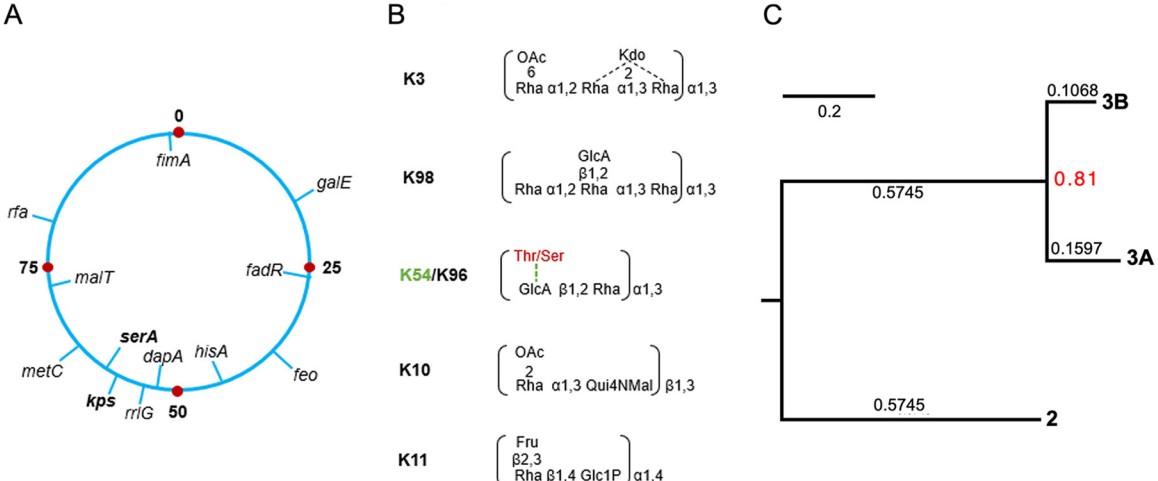

**FIG 1** *E. coli* group 3 capsule loci and known repeat-unit structures. (A) The *kps* locus in the genetic map of the *E. coli* chromosome (drawn based on the linkage maps by B. J. Bachmann, 61). (B) Known group 3 capsular structures. (C) Relative evolutionary divergence between subgroups 3A and 3B at family level. Fig. 1B and C were redrawn from reference 23. OAc, *O*-acetyl group; Rha, rhamnose; KDO, keto-deoxyoctulosonate; GlcA, glucuronic acid; Thr, threonine; Ser, serine; Fru, fructose; Qui4NMal, 4,6-dideoxy-4-malonylaminoglucose, and Glc, glucose. Note that theonine/serine and K54 were highlighted in red to indicate that the modification is K54 exclusive.

(2). Cell encapsulation, plus the structural heterogeneity of capsules, protects the bacterium against various environmental stresses (3–6).

Capsular polysaccharide is instrumental in host-pathogen and predator-prey interactions with phages (7). While some capsules in *E. coli*, such as K1 and K92, were found to interrupt the host complement cascade to escape innate immunity, encapsulation by other capsule types, such as K5, does not confer serum resistance (8, 9). Several additional studies also revealed cases where the production of a surface capsule could enhance phagocytosis by the host (8, 9). The role of capsules in phage interactions is also complex. Some capsules are protective against phage attacks by providing an effective physical barrier, but others often constitute the primary receptor for the binding of the phage tail spikes (10–14).

The biosynthetic machineries for different capsular polysaccharides are also complex. Four bacterial polysaccharide pathways are known, but the most exploited biosynthetic pathways are the Wzx/Wzy-dependent and ATP-binding-cassette (ABC) transporter pathways (1, 15–21). Relative to other bacterial species, the capsular polysaccharide pathways in *E. coli* are arguably the most complex. First, *E. coli* employs both major pathways (Wzx/Wzy and ABC pathways) for capsular biosynthesis (1); second, these capsular biosynthetic pathways could be further divided into four distinct groups (groups 1 to 4) based on differences in the biosynthetic process, gene cluster arrangement and mode of regulation control (1). The chromosomal loci harboring the capsular gene clusters (*kps*) also differ. Group 1 and 4 *kps* gene clusters are localized to the *galF/gnd* locus, while several group 2 and 3 *kps* gene clusters were mapped near the *serA* locus through Southern blot analysis (1) (Fig. 1A).

Group 2 and 3 capsules are made by the ABC transporter pathway (1). While group 2 capsules have been well studied, comparatively little is known about group 3 structure and function, with most of our understanding being inferred from biochemical and structural characterizations of group 2 (1). The ABC transporter pathway involves the synthesis of complete polysaccharide strands linked to a short poly-Kdo linker on a lipid carrier, *lyso*-phosphatidylglycerol, at the cytosolic face of the cytoplasmic membrane (22). The termination steps in biosynthesis are coupled to the transport of the polysaccharide to the periplasmic face, then to the cell surface by an ABC transporter complex consisting of at least four proteins KpsDEMT (1). Like all genetically defined group 2 *kps* clusters, group 3 clusters share a three-region gene organization format,

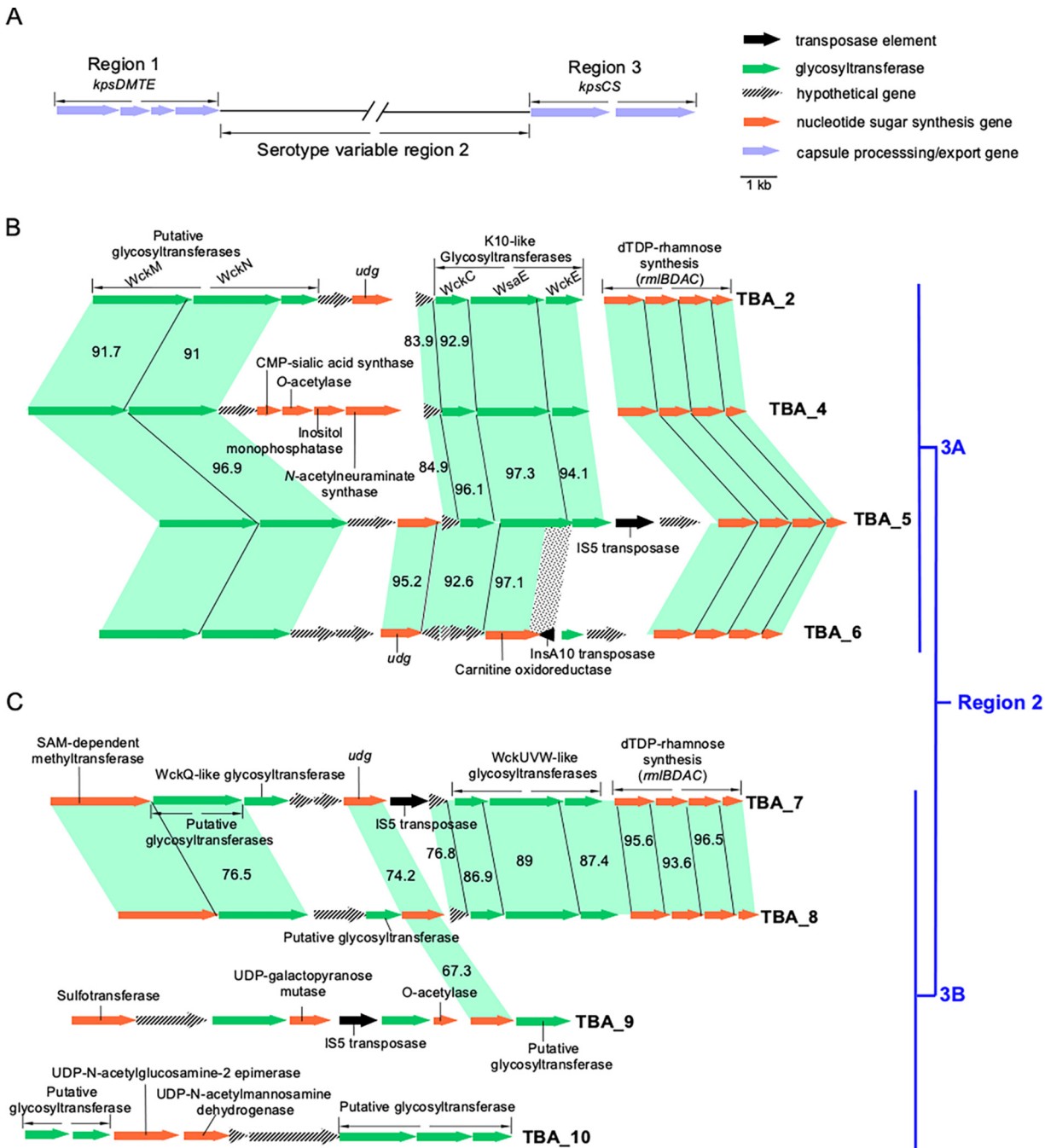

**FIG 2** Newly identified *E. coli* K-antigen group 3 gene clusters. (A) Organization of group 3 *kps* gene cluster. (B) Region 2 of three new subgroup 3A (and previously identified TBA_2) *kps* clusters. (C) Region 2 of four new subgroup 3B *kps* clusters. Genes were colored accordingly to putative functions. The identity (%) between related genes was given, except for those with identity above 98%. The dotted shade between TBA_5 and TBA_6 indicates the region corresponding to the 3′ end of the *wsaE* gene sharply displaced by the insertion of an unrelated transposase gene.

except they encode the conserved ABC transporter genes (*kpsDMTE*) in region 1 (instead of region 3 in group 2), and the *kpsCS* genes for Kdo oligosaccharide linker assembly are in region 3 (instead of region 1 in group 2) (see Fig. 2A). In both groups, the conserved regions 1 and 3 flank the variable central region 2 that houses serotype-specific genes dedicated to the synthesis of nucleoside sugars (and other structural components) and glycosyltransferases (GTs) (23). We recently uncovered unexpectedly high sequence divergence in the "conserved" regions 1 and 3 of group 3 *kps* clusters, leading to their further division into subgroup 3A and 3B (Fig. 1C) (23). Given their

long evolutionary history, we predict that further variation must exist within group 3 capsules that we set out to uncover.

Novel K serotypes are frequently reported for many species, such as *Klebsiella pneumoniae* (24). This is not the case for *E. coli*, where the degree of capsular diversity is commonly believed to be close to final (25–27). However, two recent independent studies suggest otherwise (23, 28). First, Nanayakkara et al. reported that some *E. coli* strains from phylogroups A, B1, and C had acquired *Klebsiella* capsules at the *galF/udg* locus reminiscent of group 1 capsules in *E. coli* (28). Our recent genetic analysis of group 3 capsules also supports that *kps* diversity in *E. coli* is underestimated (23). Only six group 3 capsular serotypes were recognized in *E. coli* until very recently (2). While their structures were elucidated in the 1990's (29–33) (shown in Fig. 1B), their genetic loci were uncharacterized until recently, leaving only K3 and K98 to be genetically determined (23, 34). Our analysis also identified three novel group 3 gene clusters from publicly available *E. coli* genomes that could not be assigned to either K3 or K98, demonstrating that the total number of group 3 *kps* clusters exceeds those previously described (23).

Here, we exploited the compact and genetically defined group 3 *kps* clusters to assess the degree to which novel *kps* types circulating in *E. coli* have been overlooked. We report the discovery of seven new *kps* gene clusters in *E. coli*. Rather than taking a phylogenetic approach, we used the existing group 3 sequences to establish the framework explaining how the diversity in related *E. coli* group 3 *kps* clusters may have evolved. Remarkably, subgroup 3B *kps* gene clusters, but not 3A, are mainly present in plasmids. This contradicts conventional knowledge stating that all group 3 clusters are proximal to the *serA* gene on the *E. coli* chromosome. There is also clear evidence for a set of related 3B clusters being shared between multidrug-resistant (MDR) *E. coli* strains and other Enterobacteriaceae family members. Given the vital role of capsular polysaccharides in phage predation and interactions, it is critical to characterize and monitor the dynamics of change for *kps* clusters in pathogenic *E. coli* strains.

## RESULTS

**Identification of seven novel group 3 *kps* gene clusters.** GenBank survey of all published *E. coli* sequences using *kps* conserved regions 1 and 3 from K10 and K11 (representatives of subgroups 3A and 3B, respectively) revealed seven new group 3 *kps* gene clusters carrying previously unreported region 2 sequences (Fig. 2B and C). Due to the lack of sufficient information to assign new gene clusters to the remaining known structures of either K3 or K98, we continue to use the temporary to-be-assigned (TBA) nomenclature adopted in our earlier work (23). We propose that these seven gene clusters, TBA_4 to TBA_10, code to synthesize novel capsule serotypes. Note that whether these gene clusters are functional needs further experimental validations. For accessions and precise localization of genomic and plasmid sequences, see Table S1.

Our previous analysis revealed a block of strikingly conserved genes encoding enzymes for dTDP-Rha biosynthesis (*rmlBDAC* genes) in the serotype-specific region 2 of all group 3 gene clusters, except K11 (subgroup 3B). Three conserved GT genes were also identified immediately upstream of the *rml* genes (*wckC-wsaE-wckE* or *wckI-wsaE-wckE* in 3A clusters and *wckUVW* in 3B clusters). This 3-GT block was proposed for synthesizing a complex oligosaccharide linker (23). In the present study, five of the seven new *kps* gene clusters (TBA_4 to TBA_8) also contained the *rmlBDAC* genes (Fig. 2B and C). In all cases but TBA_6, the conserved 3-GT block is identified upstream of the *rml* genes (Fig. 2B and C). The tight coexistence with *rml* genes with the 3-GT block and the finding that the TBA_6 cluster only contains the *rml* genes suggests that the proposed linker likely contains Rha.

A consensus feature of group 3 *kps* gene clusters is that they are located on a chromosomal locus close to the *serA* gene (Fig. 1A) (1). Consistent with this, all eight subgroup 3A clusters were localized within 20 to 150 kb upstream of the *serA* gene and downstream of the *rrfG* gene (Fig. 3A). However, all 3B clusters, except TBA_3 (Fig. 3A), were present in plasmid sequences instead of mapping to the *serA* chromosomal locus of *E. coli* (Fig. 3B). Note

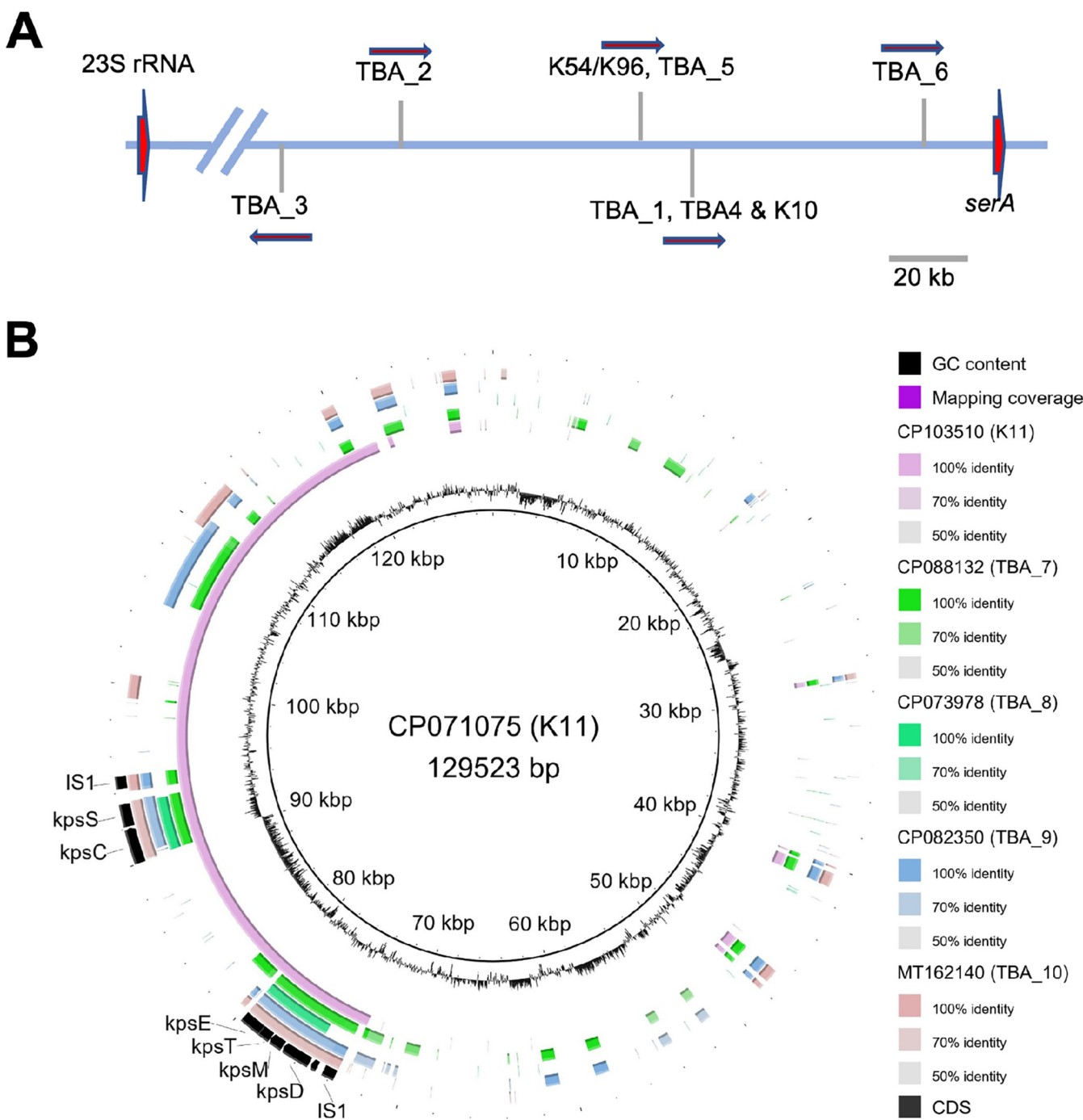

**FIG 3** Localization of group 3 *kps* gene clusters. (A) Subgroup 3A clusters and the only 3B cluster TBA_3 are mapped to a position on the *E. coli* chromosome flanked distantly by the 23S rRNA *rrfG* and *serA* genes. (B) Subgroup 3B clusters were detected on plasmids. The full plasmid sequence carrying the K11 *kps* gene cluster (CP103510.1) was compared against other 3B plasmids. The innermost ring shows GC content, following by ring showing sequential comparisons to another K11 plasmid, and then plasmids carrying TBA_7, TBA_8, TBA_9, and TBA_10 *kps* clusters against CP103510.1 (K11).

that K11, TBA_7, TBA_9, and TBA_10 clusters are flanked by conserved IS1 transposase elements that may be instrumental in mobilizing these *kps* loci (Fig. 3B).

Region 2 typically encodes GTs and synthetases required to make specialized nucleotide sugars and precursors for other structural constituents. In this respect, the gene content of region 2 in the new clusters provided a reasonable way to assess the level of complexity for the repeat unit for which structural information remains unknown. Out of the seven newly identified clusters, all but TBA_4 and TBA_10 carry an additional copy of the *udg* gene for UDP-glucuronic acid (GlcA) biosynthesis, suggesting

the five group 3 clusters likely produce an acidic capsule (Fig. 2). Other synthetases potentially encode structural component precursors, CMP-sialic acid, myco-inositol, CMP-*N*-acetylneuraminic acid (Neu5Ac), UDP-galactofuranose (Gal*f*), UDP-*N*-acetyl-mannosamine (ManNAc), UDP-*N*-acetylmannosaminuronic acid (ManNAcA), as well as modification genes encoding *O*-acetylase, methyltransferase, and sulfotransferase (Fig. 2). Two to five GT genes were identified in region 2 in each of these seven clusters (Fig. 2), but we currently lack structural data to assign any speculative functions.

**Several subgroup 3A gene clusters have likely originated by diversification from ancestral forms using shared central region genes.** Although the conserved regions 1 and 3 may be the primary recombination spot for diversity in the *kps* locus, it is hard to speculate their history confidently, given that currently observed cluster forms are likely derived from intermediate forms no longer in existence. However, the central regions (region 2) of the newly identified TBA_4, TBA_5, and TBA_6 *kps* clusters are very similar to that from TBA_2 (Fig. 2). Aside from sharing almost identical *rml* genes, two GT genes immediately after the conserved region 1 were prevalent in all three *kps* clusters and shared ∼ 92% nucleotide identity to *wckM* and *wckN* from TBA_2 (CA696_004240 and CA696_004245 respectively, in CP023142.1). These traits suggest that diversification may have occurred through recombination at these shared region 2 genes. DualBrothers was used initially to detect potential recombination events between these three gene clusters (35). Nonetheless, genetic changes could not be inferred, presumably because of the large, gapped regions in the variable central region; therefore, substitution patterns were instead employed to investigate the history of the *kps* clusters (Fig. 4).

A cluster of substitutions was observed at the 3′ end of the *wckN* gene between TBA_4, TBA_5, and TBA_6 (Fig. 4A). Furthermore, SNP density analysis revealed a sharp increase (from less than 10% to 37%) in the last 100 bp region of the GT gene (Fig. 4B). A central region is shared between TBA_4 and TBA_6 to TBA_5 (Fig. 2B). TBA_5 and TBA_6 clusters host an almost identical *udg* gene (95.2%) similar to the orthologue in TBA_2 (92.2% and 91.9% identity, respectively), while this gene is absent in TBA_4. In TBA_6, although the genes after *udg* are annotated as four hypothetical genes and a putative carnitine oxidoreductase gene, they are homologous to a related hypothetical gene, and *wckC-wsaE* proposed to be involved in linker synthesis shared by diverse group 3.

Remarkably, a similar but reversed SNP density pattern in *wckM/wckN* was observed in the shared *udg* gene between TBA_5 and TBA_6, with 25% substitutions found in the first 100 bp of the gene that then quickly declined to below 5% (Fig. 4C). It seems highly likely that *wckN* and *udg* (for TBA_5 and TBA_6) constitute part of the recombination hot spot that gave rise to TBA_4, TBA_5, and TBA_6 clusters. However, the relationship of TBA_4, TBA_5, and TBA_6 is further complicated by additional differences observed further down the 3′ end and just before the *rml* genes (Fig. 2B). These variations might be related to transposase activities. The evidence is particularly strong for TBA_6; most of the putative carnitine oxidoreductase sequence resembles that of the *wsaE* gene (97.1%), and an InsA10 transposase gene displaces the 3′ end sequence from the reverse complement direction (Fig. 2B).

**Plasmid-borne TBA_7 and TBA_8 clusters are highly related to the chromo-somal TBA_3 *kps* cluster.** Three 3B clusters, including TBA_3, TBA_7, and TBA_8, shared similar gene content in central region 2. Thus, we determine the SNP densities in their shared/conserved loci to investigate common ancestry. Region 2 in TBA_3 and TBA_8 differed only in the presence of two distinctive genes (Fig. 5A). Although the first two region 2 genes were almost identical between the two clusters, we identified 87 nucleotide substitutions condensed in the last 664 bp of the second region 2 gene (locus tag: BHT24_16990 for TBA _3) (Fig. 5D). Interestingly, continuous substitutions involving four spikes of high SNP densities between TBA_3 and TBA_8 were observed from *udg* through to *wckU* but this pattern suddenly disappeared in the subsequent *wckV* gene (Fig. 5E). Based on the observed substitution patterns, we propose recombination occurred at BHT24_16990 in TBA_3 (or equivalent locus in TBA_8) and the *udg-wckV* gene. In this case, the several spikes of SNP density may represent footprints of multiple previous recombination events in this locus.

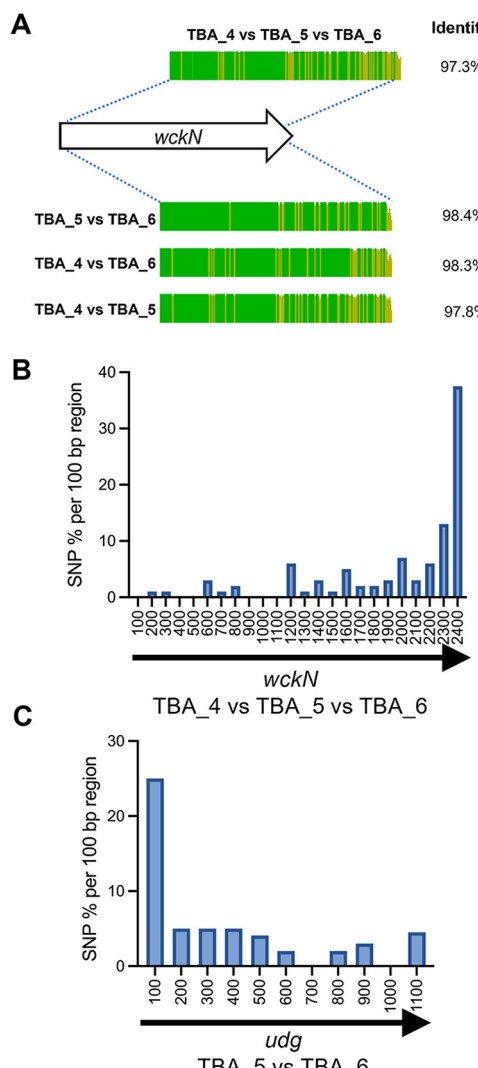

**FIG 4** SNP distribution frequencies reveal putative recombination points between TBA_4, TBA_5, and TBA_6 clusters. (A) The ClustalW aligned identity map of the *wckN* gene from TBA_4, TBA_5, and TBA_6. Note that the score identity from TBA_4 versus TBA_5 versus TBA_6 was derived from multiple sequence alignment while the rest were identity scores from pairwise comparisons. (B) SNP density of the *wckN* gene sequences from TBA_4/5/6 clusters over 100 bp regions. (C) SNP density of the *udg* gene between TBA_5 and TBA_6. The SNP density was calculated as the percentage of nucleotide substitutions across regions of 100 bp length. Note that the numerical values at the *x* axis in panels B and C refers to the positions of gene sequences being compared.

TBA_7 is also very similar to TBA_3 and TBA_8. However, many shared genes in region 2, including the 3-GT block and several *rml* genes, contained strings of contiguous substitutions compared to TBA_3 and TBA_8 (Fig. 5A). Interestingly, this substitution pattern was sharply halted at the first region 2 gene immediately after region 1, a SAM-dependent methyltransferase gene. Most nucleotide substitutions (41 nucleotides) identified in the shared methyltransferase gene between TBA_3 and TBA_7 were clustered to the last 263 bp of the coding sequence (Fig. 5B). The dTDP Rha biosynthesis genes are remarkably conserved, and cases where they showed varied substitutional patterns often reflects recombination history (36, 37), and was discussed in detail by Reeves et al. (38). Interestingly, although *rmlBAD* genes are almost all identical between TBA_3 and TBA_8 (>99.7% identity), only the *rmlC* gene just before region 3 is identical between the three clusters (Fig. 5C). The overall pattern suggests that *rmlC* served as another recombination junction. It appears that region 2 in TBA_3 (and TBA_8) has a separate lineage of origin from that of TBA_7.

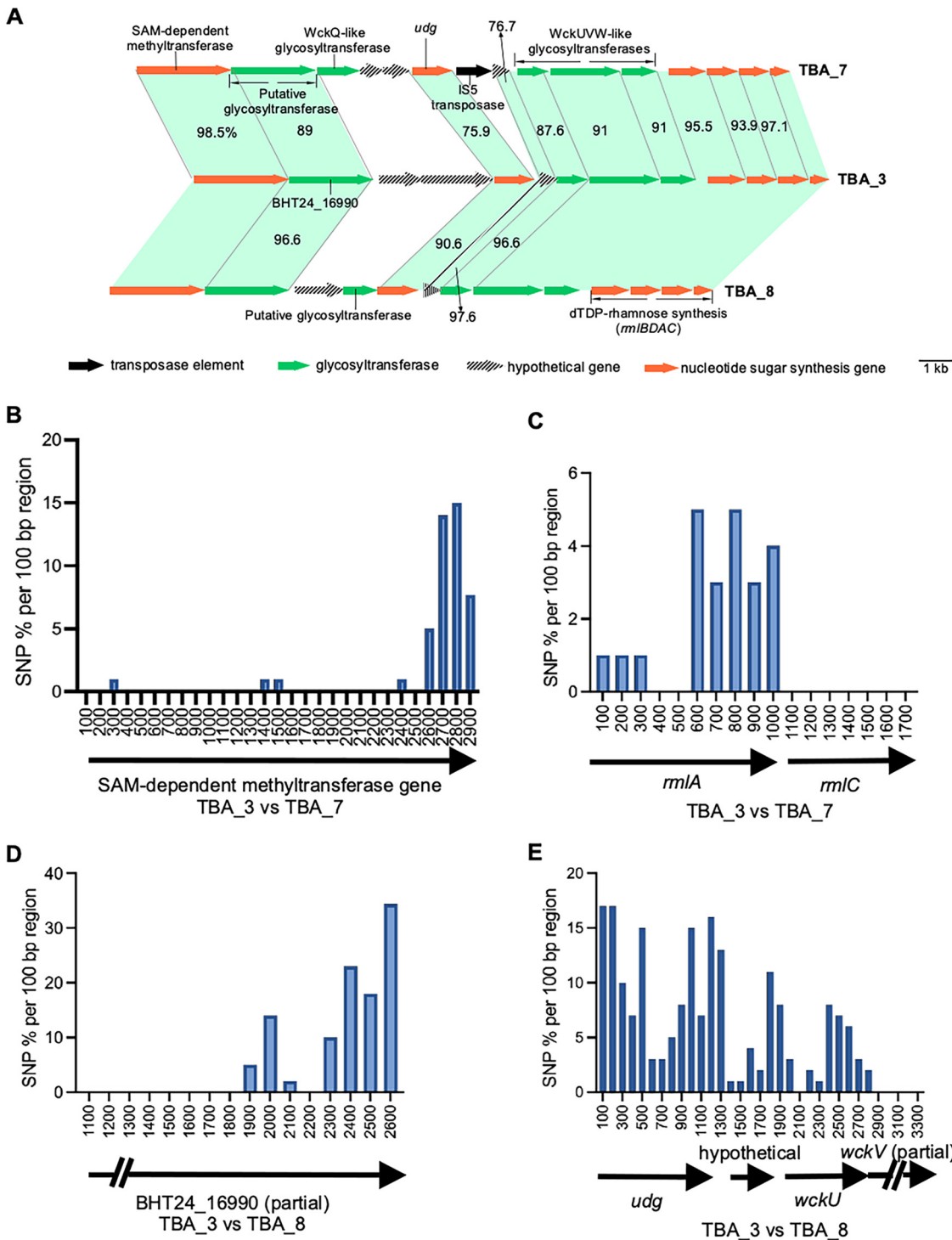

**FIG 5** SNP analysis of shared genes between TBA_3, TBA_7, and TBA_8 *kps* clusters. (A) Region 2 of TBA_7 and TBA_8 clusters from plasmids are highly related to the chromosomal TBA_3 cluster (identities were derived from pairwise comparisons with each related gene from TBA_7 and TBA_8 to TBA_3). Where no numbers are shown, it denotes genes with a pairwise identity of > 98% to reference TBA_3 sequences. (B and C) SNP density maps for related genes from TBA_7 and TBA_3. (D and E) SNP density maps of related genes from TBA_8 and TBA_3. The SNP density was calculated as the percentage of nucleotide substitutions present across regions of 100 bp length. Note that the numerical values at the *x* axis in panels B, C, D, and E refers to the positions of gene sequences being compared.

**Insights into the evolution of subgroup 3B *kps* clusters from other Enterobacteriaceae species.** We previously posited that subgroups 3A and 3B have been diverging for an enormous amount of time before appearing in *E. coli* (Fig. 1C) (23). The finding that 3B clusters are often present on plasmids suggests they are a

**Table 1** Genome and plasmid sequences from non-*E. coli* species and strains encoding subgroup 3B-like *kps* clusters[a]

| Bacteria | Location | Accession No. | Position (Start to End) | Region 1 % NT.ID [b] (% coverage) | Region 3 % NT.ID (% coverage) | Three-GT block | *rmlBCAD* |
|---|---|---|---|---|---|---|---|
| *Leclercia* sp. G3L | C | CP088265.1 | 537470 to 568288 | 97.06 (100) | 86.19 (96) | No | Yes |
| *L. adecarboxylata* strain L21 | P | MN423365.1 | 7593 to 31994 | 97.99 (100) | 98.83 (92) | Yes | Yes |
| *Enterobacter hormaechei* strain RHBSTW-00040 | P | CP058177.1 | 9006 to 37888 | 93.03 (99) | 96.78 (96) | Yes | Yes |
| *L. adecarboxylata* strain kcgeb_e1 | C | CP098325.1 | 592378 to 569918 | 90.97 (100) | 88.00 (96) | Yes | Yes |
| *L. adecarboxylata* strain 707804 | C | CP049980.1 | 186361 to 160033 | 89.46 (100) | 86.77 (96) | Yes | Yes |
| *E. hormaechei* strain RHBSTW-00059 | P | CP058169.1 | 5165 to 30059 | 89.15 (100) | 97.96 (96) | Yes | Yes |
| *E. asburiae* strain CAV1043 | P | CP011590.1 | 83480 to 9327 | 88.07 (99) | 97.99 (96) | Yes | Yes |
| *Leclercia* sp. LSNIH1 | C | CP026387.1 | 1804966 to 1830886 | 85.34 (99) | 86.72 (96) | Yes | Yes |
| *Leclercia* sp. W17 | C | CP031101.1 | 4199260 to 4215744 | 85.45 (100) | 95.85 (96) | No | No |
| *Citrobacter freundii* complex sp. CFNIH4 | C | CP026231.1 | 3594205 to 3618799 | 85.48 (98) | 85.55 (96) | Yes | Yes |
| *L. adecarboxylata* strain Z96-1 | C | CP040889.1 | 2715785 to 2688935 | 85.03 (100) | 88.98 (96) | Yes | Yes |
| *L. adecarboxylata* strain SH19PE29 | C | CP087280.1 | 576686 to 554506 | 85.2 (98) | 86.90 (96) | Yes | Yes |
| *E. roggenkampii* strain RHBSTW-00002 | P | CP058198.1 | 5165 to 34689 | 99.73 (100) | 97.99 (96) | Yes | Yes |

[a]Only those with nucleotide identity ≥ 85% to the *E. coli* group 3 K11 sequence were listed.
[b]NT.ID, nucleotide identity; P, plasmid; C, chromosome.

relatively recent acquisition by *E. coli*. Bacterial surface polysaccharide gene clusters typically have a lower GC content (≤ 40%) than the genome average of *E. coli*. This atypical GC % provides a strong basis to indicate that the clusters were acquired by lateral transfer and allows tightly controlled expression in the host species through rare codons and H-NS silencing (39). All subgroup 3A clusters maintain a consistent GC content at 40% to 43%, in line with that expectation. In contrast, the GC content of subgroup 3B gene clusters is considerably higher than that of 3A clusters (between 46.3% and 52.2%, except TBA_9 at 44.3% due to a 39.8% GC region 2), including TBA_3, the only 3B cluster found on the chromosome of *E. coli* (Table S1). Higher GC content areas were particularly observed across conserved *kps* regions 1 (~ 50%) and 3 (≥ 56%) and supports an independently evolving ancestry of 3B from subgroup 3A clusters.

Extended sequence searches among non-*E. coli* sequences present in GenBank using the conserved regions 1 and 3 from the K11 cluster (MG736913.1) revealed high identity chromosomal and plasmid sequence hits in *Leclercia*, *Citrobacter*, and *Enterobacter* species (Table 1). The non-*E. coli* chromosomal sequence hits in particular hold interest, as they may reveal the origins of subgroup 3B clusters. Interestingly, region 1 from K11 had the highest identity to *Leclercia* sp. G3L (CP088265.1), although much less identity is shared between their region 3 sequences (86.19%). Region 3 of K11 resembled that of *Leclercia* sp. W17 (CP031101.1) at 95.86%, but this is not the case across region 1. To investigate this further, phylogenetic trees were constructed for region 1 and region 3, using sequences from all six 3B clusters from *E. coli* and the eight chromosomal 3B-like clusters (Table 1) from *Leclercia* and *Citrobacter* (Fig. 6). It is expected that if the two *kps* regions in subgroup 3B clusters have evolved from a single progenitor, the two trees would share similar topologies. Indeed, this is the case for clusters TBA_3, TBA_7, and TBA_8, suggesting they have likely diverged through horizontal gene transfer using shared genes within region 2 (Fig. 5). This finding supports the use of tree topology to infer the origins of *kps* regions considered to be conserved (Fig. 6A and B).

Clusters TBA_9 and TBA_10 formed a separate branch shared in both trees (Fig. 6). However, while their region 1 sequences grouped with *Leclercia* sp. W17 (CP031101.1) (Fig. 6A), region 3 sequences were divergent from W17, and constitute members in two distinctive major clades (Fig. 6B). A similar pattern was observed with K11 in that

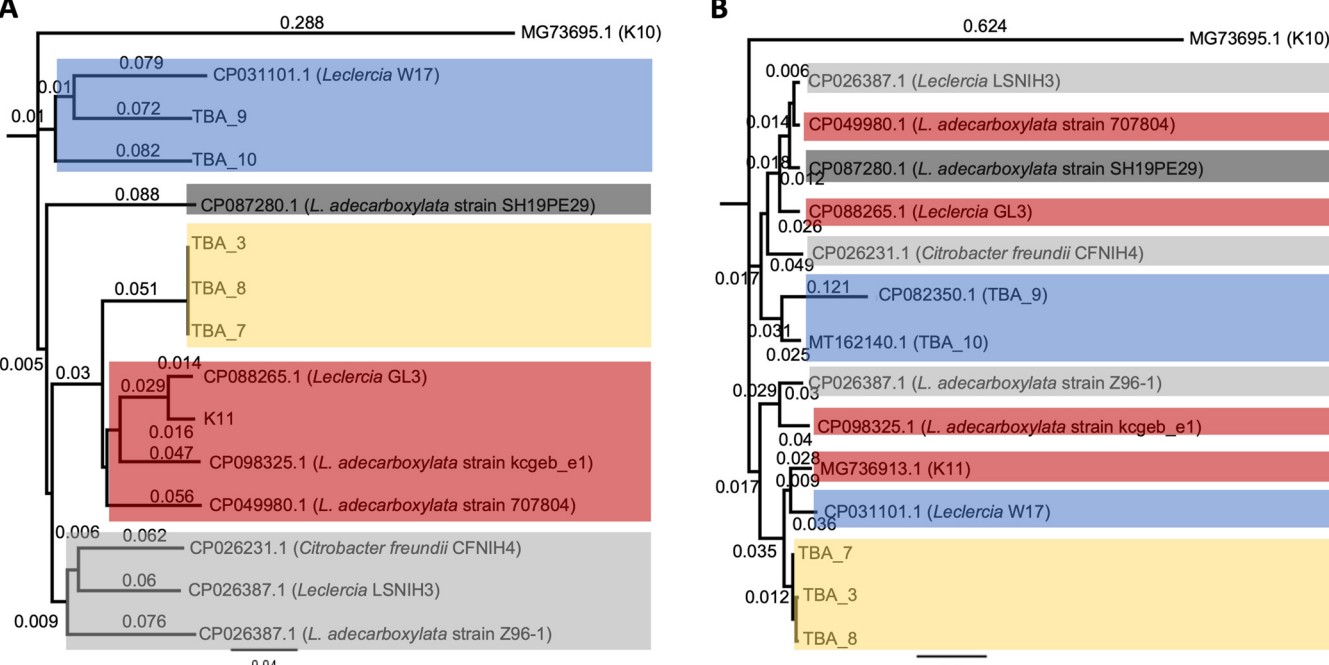

**FIG 6** Phylogenetic trees constructed with region 1 and region 3 sequences from *E. coli* subgroup 3B *kps* clusters and 3B-like clusters present in other Enterobacteriaceae species. (A) Region 1 phylogenetic tree. (B) Region 3 phylogenetic tree. Regions 1 and 3 from K10 (subgroup 3A) were used as the outgroup in A and B, respectively. Neighbor-joining trees were constructed with Geneious Tree Builder using ClustalW alignments of the nucleotide sequences of regions 1 and 3 from group 3 *kps* clusters and 3B-like clusters of other Enterobacteriaceae species with a bootstrap value of 10,000 (inbuilt into Geneious 8.1.9, https://www.geneious.com). Each major clade in panel A tree (region 1) was color-coded, and sequences within these clades were assigned the same color in panel B tree (region 3) to illustrate that the two regions likely have separate origins.

its region 1 and region 3 sequences were grouped separately between the two phylogenetic trees (Fig. 6). Overall, the incongruent trees suggest that regions 1 and 3 in subgroup 3B *kps* clusters have evolved from multiple ancestries or that each region was separately acquired by two independent horizontal gene transfers. We propose that the subgroup 3B clusters likely evolved through independent recombination events across different Enterobacteriaceae species before entering *E. coli* by plasmid-mediated transfers, and TBA_3 demonstrates how this could be captured into the typical chromosomal *kps* locus for stable vertical transfer.

**Variations in group 3 *kps* clusters within dominant *E. coli* clonal lineages.** To examine the distribution of group 3 *kps* clusters across different *E. coli* lineages, the 47 *E. coli* genomes found to encode a group 3 *kps* cluster at the *serA* locus were typed according to the Max Planck Institute Scheme (40) (Table S2) and allocated into 12 sequence types (STs). The lineage with the highest frequency of group 3 *kps* clusters was the uropathogenic ST69 lineage (13/47 genomes; Fig. 7A). Of those, the majority encoded a K54/K96 cluster (53.8%), followed by K10 (23.1%) (Fig. 7B). Clusters TBA_2, TBA_3, and TBA_4 were also found in single ST69 isolates, demonstrating diversity in *kps* types in this dominant *E. coli* lineage (Fig. 7B). Two additional STs represented by more than five genomes (ST405 and ST1312) also harbored more than one group 3 *kps* types, with TBA_5 and K54/K96 being the major *kps* types, respectively (Fig. 7B). Overall, group 3 *kps* clusters are not confined to a specific clonal lineage, as the presence of two to four distinct group 3 clusters was observed in more than one ST.

## DISCUSSION

The current estimate of 80 capsule forms in *E. coli* is commonly believed to be close to final (25, 26). However, we previously identified surplus gene clusters that could not be accommodated by the known group 3 capsule scheme (23). This led us to speculate the possibility of unexpected diversity among group 3 *kps* gene clusters in *E. coli*, and we confirmed this is the case here. Group 3 clusters are surprisingly diverse. Aside from

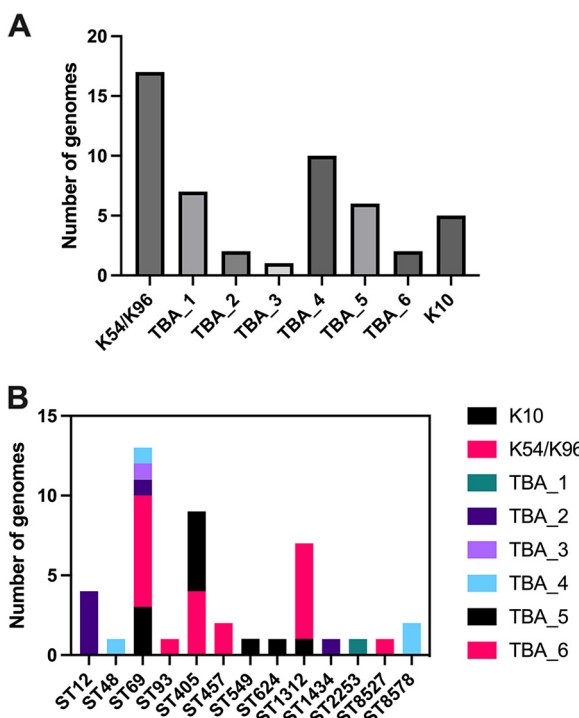

**FIG 7** Frequency of chromosomal group 3 *kps* gene clusters in different *E. coli* sequence types. (A) Histogram of *kps* types across 47 *E. coli* genomes encoding a group 3 capsule. (B) Clonal lineage variations in *kps* types: Histogram of *kps* types per *E. coli* sequence type from 47 genomes encoding a group 3 capsule.

the K10, K11, and K54/K96 that we could associate with the original scheme, we identified 10 TBA *kps* gene clusters evenly allocated to the two divergent subgroups. It appears that wide group 3 *kps* diversity is circulating in the species, but this has gone undetected as the K typing scheme has been long abandoned. A gross underestimation of diversity may also be true for other capsule groups in *E. coli*.

The *kps* locus position of group 3 capsules in *E. coli* was first reported for K10 and K54 through Southern Blot analysis, before its division from Group 2 (41–43). Subsequent works validated shared locus position in several other group 2 encoding strains before this localization was adopted as common knowledge for both groups (1, 44). Although this holds true for all seven 3A *kps* clusters (proximal to the chromosomal *serA* gene), we have shown that all 3B *kps* clusters (except TBA_3) are located on plasmids (Fig. 3). Notably, while we could identify 3B-like gene clusters in other members within Enterobacteriaceae, no closely related clusters were identified for 3A. For one, this finding highlights the possibility that 3A clusters have been evolving in the species for a very long time, while 3B clusters are a relatively recent acquisition in *E. coli*. Overall, these results provide further support that 3A and 3B subgroups had diverged a long time ago (Fig. 1C) (23).

The primary driver for structural heterogeneity of capsular polysaccharides in group 2 and 3 capsules is the variations within their central region 2. However, there is very little indication of how such variation is generated, with typically the relatively conserved regions 1 and 3 thought to constitute the primary recombination arms to introduce altered polysaccharide forms. We find that region 2 in several of the existing *E. coli* group 3 *kps* forms were strikingly similar, and there is substantial genetic evidence that these gene clusters have diverged through recent recombination events (Fig. 2, 4, and 5). One particular highlight is the *wsaE* GT gene in TBA_5 (shared gene by most subgroup 3A). In the TBA_6 cluster, the original *wsaE* gene was interrupted by an InsA10 transpose, and the new annotation predicted it as a carnitine oxidoreductase gene (Fig. 2A). There is a sharp break from no identity to high identity at the junction into the carnitine oxidoreductase gene. It is worth mentioning that these observed sequences may be very different from the original recombination product, as there

would be changes driven by selection to refine the function of the disrupted *wsaE* gene. It is interesting that for the Wzx/Wzy pathway, cases in which only subtle differences exist between polysaccharide gene clusters are relatively rare, given that the evolution of a new form in this pathway required the acquisition of a Wzx and Wzy that could adapt to prevent cell lysis (19, 21). We are unaware of any restrictions to the evolution of surface polysaccharide forms produced by the ABC transporter pathway; as such, perhaps the trade-off is little, and very intricate changes are sufficient to evolve novel polysaccharide forms.

Regions 1 and 3 for 3B *kps* clusters are particularly interesting because we could allocate them to different recent ancestries shared by other members of Enterobacteriaceae (Fig. 6). In short, our analysis provides clear evidence that *kps* clusters are recombination hot spots. Several possible paths can introduce variations in the serotype-specific region 2, shedding some light on the evolutionary trajectory of capsular biosynthesis. First, new *kps* forms may be formed by recombination events between shared genes within region 2. Second, it is also possible that an entire block of region 2 may be captured into existing *kps* clusters outside of the species based on sequence conservation of flanking regions 1 and 3. Recombinant products may also be subjected to further divergence through random genetic drift or selection pressure to introduce antigenic variation required to override existing host immunity or receptor interactions under phage predation. It is unexpected that some *kps* gene clusters can result from reassortments of supposedly conserved regions 1 and 3 sourced from different origins and, in this process, presumably acquire novel region 2 genes. In this respect, the diversification process of group 3 could harvest on the existing capsular complexities in related species.

Irrespective of what may be driving *kps* diversification, the finding of rigorous changes in capsular polysaccharides in *E. coli*, including in pathogenic clinical isolates that carry MDR genes (Tables S2 to S4), has important health implications. *E. coli* is the leading cause of mortality associated with antimicrobial resistance, accounting for over 800,000 lives globally in 2019 alone (45). Several recent studies have proposed phage therapy as a potential alternative to antibiotics against MDR *E. coli* infections (46). However, it is well-established that intricate surface encapsulation changes can affect phage binding to initiate an attack (10–14, 47, 48). In light of this, our discovery of an unexpected expansion of capsular diversity highlights the importance of reinstating the K-antigen typing scheme. This work also raises attention to the need to track capsular changes during and after administration of phage therapies to ensure their sustained safety and efficacy as standalone infection control regimes or as antibiotic adjuvants.

## MATERIALS AND METHODS

**Detection of novel group 3 *kps* gene clusters.** Known sequences of conserved region 1 (*kpsDMTE*) and region 3 (*kpsCS*) from K10 (Accession No.: MK278915) of subgroup 3A and K11 (Accession No.: MG736913) of subgroup 3B clusters were used to identify putative new group 3 clusters in published sequences with BLAST (49) and parameters restrictive to those of *E. coli*, using a cut-off 82% identity (>99% coverage). Manual curations were used to exclude sequences resembling each other (>98% identity across region 2). Briefly, the sequences were compared and analyzed with progressiveMauve for regions shared between gene clusters to yield seven new gene clusters (50). This work assigned a temporary to-be-assigned (TBA) nomenclature from Hong et al. (23) to these new gene clusters.

***kps* sequence analysis.** Genes, gene products, and regions with shared homologies identified from progressiveMauve (50) were further examined by built-in programs from the Geneious software package (Geneious 8.1.9). ClustalW Alignments were performed at default settings (cost matrix: IUB; Gap penalty: 15; Gap extend cost: 6.66) with a free end gaps (51). Recombination analysis was performed by assessment of sequences with shared homologies in regions of 100 bp in either ClustalW pairwise or multiple sequence alignments. The substitutional distributions between compared sequences were calculated to percentage values and plotted to determine SNP densities.

**Phylogenetic analysis.** The multiple sequence alignments of shared regions 1 and 3 in related subgroup 3B *kps* clusters and 3B-like clusters from other Enterobacteriaceae species were processed and separated using ClustalW performed at default settings (cost matrix: IUB; Gap penalty: 15; Gap extend cost: 6.66) with free end gaps (inbuilt into Geneious 8.1.9) (51). The alignments were then concatenated to generate the neighbor-joining phylogenetic tree using Geneious Tree Builder with a bootstrap value of 10,000 (inbuilt into Geneious 8.1.9).

**Multilocus sequence typing.** Fifty *E. coli* genomes were identified to encode a group 3 *kps* cluster mapped to the *serA* locus. The FASTA files of the GenBank sequences were downloaded, and MLST was performed according to the Max Planck Scheme with MLST 2.0 (40, 52). Forty-seven genomes (of the 50

genomes) were allocated into 12 STs, while three were disqualified due to imperfect hits (see Table S2 for detail). These data were combined with the known chromosomal group 3 capsular gene clusters to determine variations in capsular polysaccharides within a given clonal lineage (see Table S2 for accessions and relevant details of the isolates).

**Search for antimicrobial resistance genes.** The identification of antimicrobial resistance genes was conducted with ResFinder 4.1 (2022-08-08 database) (53–58).

**Visual presentation.** Visual comparisons of plasmid sequences were performed using BLAST Ring Image Generator (59), Artemis ACT (60) and Geneious 8.1.9.

## SUPPLEMENTAL MATERIAL

Supplemental material is available online only.

**SUPPLEMENTAL FILE 1**, XLSX file, 0.02 MB.

## ACKNOWLEDGMENTS

Y.H., J.Q., and M.T. were supported by the Australian Research Council (DP190101613), the National Health and Medical Research Council of Australia (GNT1144046), and the Clive and Vera Ramaciotti Foundation (Health Investment Grant 2017HIG0119). Y.H. and M.T. are also supported by funds from the Max Planck Queensland Centre at the Queensland University of Technology (QUT). Y.H. gratefully acknowledges support from the Faculty of Health Early Career Researcher Schemes of QUT. X.B.F. is a recipient of an QUT Amplify Scholarship. All funders had no role in study design, data collection and analysis, decision to publish, or preparation of the manuscript.

Y.H. contributed to conceptualization and design, data collection, data analysis and interpretation, and data presentation and supervision; M.T. supervised the study, contributed to data interpretation, and obtained the funding; J.Q. and X.B.F. contributed to data analysis; Y.H. drafted the manuscript; Y.H. and M.T. substantially revised the manuscript. All authors agreed to the publication of the manuscript.

We declare no competing interests.

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
