## [Reviewer comments · Microbiology Spectrum]

Microbiology Spectrum

Extensive diversity in *Escherichia coli* Group 3 capsules is driven by recombination and plasmid transfer from multiple species

Yaoqin Hong, Jilong Qin, Xavier Forga, and Makrina Totsika

Corresponding Author(s): Yaoqin Hong, Queensland University of Technology

Review Timeline:

Submission Date:	April 4, 2023
Editorial Decision:	May 8, 2023
Revision Received:	May 18, 2023
Editorial Decision:	June 2, 2023
Revision Received:	June 2, 2023
Accepted:	June 4, 2023

Editor: Xiaoyu Tang

Reviewer(s): The reviewers have opted to remain anonymous.

Transaction Report:

DOI: <https://doi.org/10.1128/spectrum.01432-23>

May 8, 2023

Dr. Yaoqin Hong
Queensland University of Technology
School of Biomedical Sciences
QUT-QIMR, 300 Herston Road
Brisbane City, QLD 4006
Australia

Re: Spectrum01432-23 (**Extensive diversity in *Escherichia coli* Group 3 capsules is driven by recombination and plasmid transfer from multiple species**)

Dear Dr. Yaoqin Hong:

Link Not Available

Sincerely,

Xiaoyu Tang

Journals Department
Reviewer comments:

Reviewer #1 (Comments for the Author):

This paper identified seven novel Group 3 kps gene clusters in *E. coli*, which fall into subgroups 3A and 3B. Remarkably, the most of 3B clusters were found on plasmids rather than on chromosome. Further genetic analysis and comparison disclosed the Group 3 clusters are closely related. The diversification of kps clusters may occurred through recombination at shared central region genes and plasmid transfer between multiple species including multidrug-resistant *E. coli* strains and other Enterobacteriaceae family members. This paper highlighted the importance to characterize and monitor the change for kps clusters in pathogenic *E. coli* strains. These findings will expand our understanding of the evolution of *E. coli* capsules. Much of

the data is interesting and well presented. This manuscript is well-written.

Minor comments:

1. Fig 2A. The glycosyltransferase should be marked as green color.
2. Fig 3B. It is clearer if TBA_7 and TBA_8 were displayed in different colors.
3. Fig S1 can be deleted since it is the same as Fig 2 in reference 23.
4. Table S1, is the stop (kpcS) right?

Reviewer #2 (Comments for the Author):

This paper analysed the publicly available E. coli genomes to identify seven novel group 3 polysaccharide biosynthesis gene clusters. Interestingly, a number of these new capsule loci were on plasmids, rather than on the chromosome. Unfortunately, there is no structural analysis of any of the novel capsule types nor analysis of whether the IS1 elements surrounding the plasmid 3B sequences were active. However, the bioinformatic analysis presented expands our understanding of capsule synthesis in E. coli and suggests evolutionary perspectives of capsule switching.

Main points:

1. Line 283; "TBA_4, TBA_5 and TBA_6 kps clusters are very similar to that from TBA_2 (Fig. 2). TBA_2 be shown in Figure 2. The lack of TBA_3 is probably OK as it is shown in Fig. 5.
2. Why are the subgroups 4, 5 and 6 not shown on Fig. 3A
3. Line 293, should this be referencing Fig. 4 (the figure with the substitution patterns).
4. I find the description of the SNP analysis shown in Fig. 5 somewhat confusing. I believe the figure is only showing region two genes, but the description starts talking about the two upstream genes, which I assume are not shown. Indeed, there is a SNP analysis of BHT24_16990 but assume this gene is just upstream of the SAM-dependent methyltransferase and is not shown in panel A. Please clarify.
5. I am not convinced the phylogeny in Fig. 6B indicates the region 3 genes of TBA_9 and _10 are closely related to C. freundii; it forms a different branch that could be equally rotated. Does the amino acid similarity show higher identity between TBA_9 and _10 and C. freundii compared to the Leclercia and L. adecaroxylata strains and, if so, at what levels?
6. Page 19, Fig. 5B is referenced three times on this page but I expect it should be Fig. 7B.
7. For strains with the 3B locus on a plasmid, is there any evidence of a partial capsule locus on the chromosome? Is there any evidence that the 3B plasmids can be found in strains with a different chromosomal capsule locus? Could this be tested experimentally by transformation to see if multiple capsule loci are tolerated?

Staff Comments:

Preparing Revision Guidelines

Please return the manuscript within 60 days; if you cannot complete the modification within this time period, please contact me. If you do not wish to modify the manuscript and prefer to submit it to another journal, please notify me of your decision immediately so that the manuscript may be formally withdrawn from consideration by Microbiology Spectrum.

This paper identified seven novel Group 3 *kps* gene clusters in *E. coli*, which fall into subgroups 3A and 3B. Remarkably, the most of 3B clusters were found on plasmids rather than on chromosome. Further genetic analysis and comparison disclosed the Group 3 clusters are closely related. The diversification of *kps* clusters may occurred through recombination at shared central region genes and plasmid transfer between multiple species including multidrug-resistant *E. coli* strains and other Enterobacteriaceae family members. This paper highlighted the importance to characterize and monitor the change for *kps* clusters in pathogenic *E. coli* strains. These findings will expand our understanding of the evolution of *E. coli* capsules. Much of the data is interesting and well presented. The manuscript is well-written.

Minor comments:

1. Fig 2A. The glycosyltransferase should be marked as green color.
2. Fig 3B. It is clearer if TBA_7 and TBA_8 were displayed in different colors.
3. Fig S1 can be deleted since it is the same as Fig 2 in reference 23.
4. Table S1, is the stop (*kpcS*) right?

Ref. Spectrum01432-23

Re: "Extensive diversity in Escherichia coli Group 3 capsules is driven by recombination and plasmid transfer from multiple species"

We thank the reviewers for the constructive feedback and the opportunity to revise our manuscript for publication in Microbiology Spectrum.

As requested, a point-by-point response to reviewer comments is provided with a copy of the revised manuscripts with track changes. We look forward to hearing back from you.

Kind regards

Yours sincerely,

Dr Yaoqin Hong (corresponding author)

Dr Makrina Totsika (corresponding author)

Response to Reviewer #1

1. Fig 2A. The glycosyltransferase should be marked as green color.

We have corrected that error as suggested.

2. Fig 3B. It is clearer if TBA_7 and TBA_8 were displayed in different colors.

The two colors were indeed distinctive. To make sure they are well distinguished from each other, we modified the figure legend to the following,

"The full plasmid sequence carrying the K11 kps gene cluster (CP103510.1) was compared against other 3B plasmids. The innermost ring shows GC content, following by ring showing sequential comparisons to another K11 plasmid, and then plasmids carrying TBA_7, TBA_8, TBA_9 and TBA_10 kps clusters against CP103510.1 (K11)."

3. Fig S1 can be deleted since it is the same as Fig 2 in reference 23.

Fig S1 was deleted as suggested.

4. Table S1, is the stop (kpcS) right?

Yes, *kpcS* is the last gene on group 3 region 3.

Response to Reviewer #2

Main points:

1. Line 283; "TBA_4, TBA_5 and TBA_6 kps clusters are very similar to that from TBA_2 (Fig. 2). TBA_2 be shown in Figure 2. The lack of TBA_3 is probably OK as it is shown in Fig. 5.

We agreed and do also feel adding TBA_2 to Fig 2 would help in understanding how they are related to TBA 4/5/6. TBA_2 had been integrated into the modified figure.

2. Why are the subgroups 4, 5 and 6 not shown on Fig. 3A

We thank the reviewer for pointing out this error. The correct panel with all chromosomal located subgroups had replaced the original incomplete panel.

3. Line 293, should this be referencing Fig. 4 (the figure with the substitution patterns).

We thank the reviewer for pointing out this error and this has been corrected.

4. I find the description of the SNP analysis shown in Fig. 5 somewhat confusing. I believe the figure is only showing region two genes, but the description starts talking about the two upstream genes, which I assume are not shown. Indeed, there is a SNP analysis of BHT24_16990 but assume this gene is just upstream of the SAM-dependent methyltransferase and is not shown in panel A. Please clarify.

We thank the reviewer for pointing out this confusion. BHT24_16990 is the second gene within the region 2 of TBA_3. We modified the description so it should be no longer confusing, "Although the first two region 2 genes were almost identical between the two clusters, we identified 87 nucleotide substitutions condensed in the last 664 bp of the second region 2 gene (locus tag: BHT24_16990 for TBA_3)". In addition, we marked the location of BHT24_16990 on Fig 5A to avoid such confusion.

5. I am not convinced the phylogeny in Fig. 6B indicates the region 3 genes of TBA_9 and _10 are closely related to *C. freundii*; it forms a different branch that could be equally rotated. Does the amino acid similarity show higher identity between TBA_9 and _10 and *C. freundii* compared to the *Leclercia* and *L. adecaroxylata* strains and, if so, at what levels?

The identity levels at amino acids also provides not additional clue and does not support a consistent higher identity between TBA_9 and _10 and *C. freundii* than others. We now agree with the comment that there is not sufficient support that the region 3 genes of TBA_9 and _10 are closely related to *C. freundii*. We rewrote the following section to remove the speculation on the relationship between the three in Fig 6B.

"Clusters TBA_9 and TBA_10 formed a separate branch shared in both trees (Fig 6). However, while their region 1 sequences grouped with *Leclercia* sp. W17 (CP031101.1) (Fig 6A), region 3 sequences were divergent from W17, and constitute members in two distinctive major clades (Fig 6B)."

6. Page 19, Fig. 5B is referenced three times on this page but I expect it should be Fig. 7B.

We thank the reviewer for pointing out this error and this has been corrected.

7. For strains with the 3B locus on a plasmid, is there any evidence of a partial capsule locus on the chromosome? Is there any evidence that the 3B plasmids can be found in strains with a different chromosomal capsule locus? Could this be tested experimentally by transformation to see if multiple capsule loci are tolerated?

We appreciate the reviewer for this insightful comment. We indeed investigated the genomes of these strains to assess the potential that they carry an original capsular locus. There are indications that these strains may be initially group 4 capsules, as they each at least carry a GfaC/GfaA gene on the polysaccharide locus. Whether or not the chromosomal loci that we observed are functional and the relative dynamics in the context of possibly making dual polysaccharides, as well as the potential advantages of doing so, remains to be assessed. Although we don't yet have the strains to do such assessments, dual polysaccharides presented on the cell surface have been reported in other species. For example, *Pseudomonas aeruginosa* is capable of concomitantly synthesising two types of LPS O antigens, with the relevant genes clustered into two distinct chromosomal loci.

June 2, 2023

Dr. Yaoqin Hong
Queensland University of Technology
School of Biomedical Sciences
QUT-QIMR, 300 Herston Road
Brisbane City, QLD 4006
Australia

Re: Spectrum01432-23R1 (**Extensive diversity in *Escherichia coli* Group 3 capsules is driven by recombination and plasmid transfer from multiple species**)

Dear Dr. Yaoqin Hong:

Thank you for submitting your manuscript to Microbiology Spectrum. As you will see your paper is very close to acceptance. Please modify the manuscript along the lines I have recommended. As these revisions are quite minor, I expect that you should be able to turn in the revised paper in less than 30 days, if not sooner. If your manuscript was reviewed, you will find the reviewers' comments below.

When submitting the revised version of your paper, please provide (1) point-by-point responses to the issues raised by the reviewers as file type "Response to Reviewers," not in your cover letter, and (2) a PDF file that indicates the changes from the original submission (by highlighting or underlining the changes) as file type "Marked Up Manuscript - For Review Only". Please use this link to submit your revised manuscript. Detailed instructions on submitting your revised paper are below.

Link Not Available

Sincerely,

Xiaoyu Tang

Reviewer comments:

Reviewer #1 (Comments for the Author):

Page 166: "to yield six new gene clusters". You mentioned seven new gene clusters.

Page 209-210, "We propose that these seven gene clusters code to synthesis novel capsule serotypes". Are you sure that the new kps clusters on plasmids have functions on making capsules? Is there any information on the capsule serotypes of the bacteria carrying Group 3B clusters plasmids? Do they produce capsules that are not from the kps cluster on chromosome? Novel serotype means producing new capsule structure. Without structure identification, it is hard to say that these gene clusters synthesize new capsule serotypes.

In Fig 2B, the spelling of "Glycosyltrasnserases" is wrong.

Table S1, is "kpcS" kpsC?

Reviewer #2 (Comments for the Author):

I am happy with the authors responses.

Preparing Revision Guidelines

Please return the manuscript within 60 days; if you cannot complete the modification within this time period, please contact me. If you do not wish to modify the manuscript and prefer to submit it to another journal, please notify me of your decision immediately so that the manuscript may be formally withdrawn from consideration by Microbiology Spectrum.

Page 166: “to yield six new gene clusters”. You mentioned seven new gene clusters.

Page 209-210, “We propose that these seven gene clusters code to synthesis novel capsule serotypes”. Are you sure that the new *kps* clusters on plasmids have functions on making capsules? Is there any information on the capsule serotypes of the bacteria carrying Group 3B clusters plasmids? Do they produce capsules that are not from the *kps* cluster on chromosome ? Novel serotype means producing new capsule structure.

Without structure identification, it is hard to say that these gene clusters synthesize new capsule serotypes.

In Fig 2B, the spelling of “Glycosyltrasnsferases” is wrong.

Table S1, is “*kpcS*” *kpsC*?

Ref. Spectrum01432-23

Re: "Extensive diversity in Escherichia coli Group 3 capsules is driven by recombination and plasmid transfer from multiple species"

We again thank the reviewers for the constructive feedback and the opportunity to revise our manuscript for publication in Microbiology Spectrum.

As requested, a point-by-point response to reviewer comments is provided with a copy of the revised manuscripts with track changes. We look forward to hearing back from you.

Kind regards

Yours sincerely,

Dr Yaoqin Hong (corresponding author)

Dr Makrina Totsika (corresponding author)

Response to Reviewer #1

1. Page 166: "to yield six new gene clusters". You mentioned seven new gene clusters.

We apologise for the error, and this has been fixed.

2. Page 209-210, "We propose that these seven gene clusters code to synthesis novel capsule serotypes". Are you sure that the new kps clusters on plasmids have functions on making capsules? Is there any information on the capsule serotypes of the bacteria carrying Group 3B clusters plasmids? Do they produce capsules that are not from the kps cluster on chromosome ? Novel serotype means producing new capsule structure. Without structure identification, it is hard to say that these gene clusters synthesize new capsule serotypes.

We agree on whether these gene clusters are functional needs experimental confirmation, and this is why we indicate that these gene clusters code novel capsules are a proposal. To clarify this point, we added a new sentence immediately after the line to reiterate that experimental validation will be required.

"We propose that these seven gene clusters, TBA_4 to TBA_10, code to synthesise novel capsule serotypes. Note that whether these gene clusters are functional needs further experimental validations."

3. In Fig 2B, the spelling of "Glycosyltrasnsferases" is wrong.

We apologise for the error, and this has been fixed.

4. Table S1, is "kpcS" kpsC?

We apologise for the error. It is indeed *kpsS*, and this has been fixed.

June 4, 2023

Dr. Yaoqin Hong
Queensland University of Technology
School of Biomedical Sciences
QUT-QIMR, 300 Herston Road
Brisbane City, QLD 4006
Australia

Re: Spectrum01432-23R2 (**Extensive diversity in *Escherichia coli* Group 3 capsules is driven by recombination and plasmid transfer from multiple species**)

Dear Dr. Yaoqin Hong:

Your manuscript has been accepted, and I am forwarding it to the ASM Journals Department for publication. You will be notified when your proofs are ready to be viewed.

Sincerely,

Xiaoyu Tang
Editor, Microbiology Spectrum
